# Knowledge Graph Construction to Facilitate Indoor Fire Emergency Evacuation

**Mingkang Da** [1,2,3], **Teng Zhong** [1,2,3,*] and **Jiaqi Huang** [1,2,3]

1   Key Laboratory of Virtual Geographic Environment (Ministry of Education of PRC), Nanjing Normal University, Nanjing 210023, China; dmksteven@njnu.edu.cn (M.D.); jqhuang@njnu.edu.cn (J.H.)
2   State Key Laboratory Cultivation Base of Geographical Environment Evolution, Nanjing Normal University, Nanjing 210023, China
3   Jiangsu Center for Collaborative Innovation in Geographical Information Resource Development and Application, Nanjing Normal University, Nanjing 210023, China
*   Correspondence: tzhong27@njnu.edu.cn

**Abstract:** Indoor fire is a sudden and frequent disaster that severely threatens the safety of indoor people worldwide. Indoor fire emergency evacuation is crucial to reducing losses involving various objects and complex relations. However, traditional studies only rely on numerical simulation, which cannot provide adequate support for decision-making in indoor fire scenarios. The knowledge graph is a knowledge base that can fully utilize massive heterogeneous data to form a sound knowledge system; however, it has not been effectively applied in the fire emergency domain. This study is a preliminary attempt to construct a knowledge graph for indoor fire emergency evacuation. We constructed the indoor fire domain ontology and proposed a four-tuple knowledge representation model. A knowledge graph was constructed with 1852 nodes and 2364 relations from 25 indoor fire events. The proposed method was tested for the case study of Henan Pingdingshan '5.25' Fire Accident in China. Results show that the proposed knowledge representation model and the corresponding knowledge graph can represent complicated indoor fire events and support indoor fire emergency evacuation.

**Keywords:** indoor fire; emergency evacuation; spatio-temporal process; domain ontology; knowledge representation model; knowledge graph

## 1. Introduction

Fire has always been a significant disaster threatening the safety of people in buildings worldwide [1,2]. According to data from the U.S. Fire Administration (USFA), building fires in the United States caused 2955 deaths and 12,425 injuries in 2021. The situation is equally severe in China. According to the Fire and Rescue Bureau of the Ministry of Emergency Management, 1,324,000 residential building fires occurred between 2012 and 2021, killing 11,634 people. Because of the suddenness and complexity of indoor fires [3], it is crucial for reasonable and effective emergency evacuation for all evacuees to minimize fire loss [4]. Therefore, there is an urgent need to study the factors influencing indoor fire emergency evacuation to improve evacuation efficiency effectively.

Traditional studies mainly rely on numerical simulation, separating the fire development process from evacuation by combining simulation results from different software [5,6]. Therefore, the mutual influence between indoor fire and emergency evacuation has not been well considered. In particular, individual behavior may change beyond expectation under the influence of fire products [7], which cannot be solved simply by setting parameters. At the same time, numerical simulation is limited to specific places, such as hotels [8], tunnels [9], and so on, which prevents it from obtaining general knowledge. On the other hand, the multi-source heterogeneous fire data from the huge number of indoor fires contains rich knowledge that has not been effectively used. A knowledge graph (KG)

can integrate, manage, and extract heterogeneous data from multiple sources at a large scale [10]. KG can fully use massive fire data, carrying out effective structural processing and extracting valuable general knowledge from it. In addition, the spatio-temporal KG is of great value for the dynamic description of the indoor fire emergency evacuation process and detailed analysis of the interaction process of various elements [11].

In this study, we constructed a KG to facilitate indoor fire emergency evacuation and designed multi-type knowledge query services for applying the KG. To achieve this goal, we first analyzed the indoor fire emergency evacuation process from multiple dimensions such as time, space, and semantics based on the perspective of geography. We explored the interaction between various elements in the process and the spatio-temporal change process. Then, according to the analysis above, we constructed the domain ontology and designed a four-tuple knowledge representation model: object, attribute, relation, and rule. We widely obtained related data from various sources based on the knowledge representation model. Then we stored the data in the graph database Neo4j, and connected entities with relations to obtain the KG. Finally, based on the obtained KG, we designed diverse knowledge query services. The query objects include historical information, evacuation results, and spatio-temporal similarity. The fire scenario's evacuation behavior and influencing factors were queried and analyzed, focusing on specific evacuee groups.

The main contributions of this research can be summarized as follows:

(1) Theoretically, we conducted a spatio-temporal process analysis to determine the basic components of indoor fire emergency evacuation. Then we constructed a domain ontology and proposed a four-tuple knowledge representation model including object, attribute, relation, and rule, which improves traditional methods in considering relations among all objects in an indoor fire emergency evacuation. To our best knowledge, this study is the first attempt to construct a KG for indoor fire emergency evacuation. The proposed method can support evacuation in other scenarios, such as earthquake evacuation [12,13], evacuation under terrorist attacks [14], and evacuation of miners [15]. The evacuation process in the above scenarios has many similarities with indoor fire emergency evacuation. For example, the rapidly changing disaster environment significantly influences evacuee behaviors, affecting the evacuation outcome. Therefore, the continuous deepening of our research will provide new ideas for evacuating the above-mentioned scenarios and have obvious development potential.

(2) On the application level, we carried out structural processing of massive indoor fire emergency evacuation data from 25 indoor fire events. We obtained 1852 nodes and 2364 relations and stored them in a graph database for further processing and analysis. In addition, the multi-type knowledge query services focus on different evacuee groups in an emergency evacuation. Considering individual differences, these services are convenient for detailed analysis of evacuation behaviors and results, especially for disabled groups with mobility difficulties. The above query services were tested in specific applications in our case study. The success rate of evacuation of different evacuee groups was reduced.

The remainder of this paper is organized as follows: Section 2 discusses the current research progress on indoor fire emergency evacuation and KG construction. Section 3 presents the method used to construct the KG of indoor fire emergency evacuation. Section 4 shows the results and relevant analysis. Discussion and Conclusion are presented in Sections 5 and 6.

## 2. Related Work

### 2.1. Indoor Fire Emergency Evacuation

Gradually, more studies have focused on indoor fire emergency evacuation in recent years because of the heavy casualties and property losses. The existing research mainly considers fire, building, and personnel. Some studies focused on the relations between fire and building objects, especially indoor fire risk assessment supported by fire simulation [16–18]. Studies mainly relied on fire simulation models like Fire Dynamics Simulator (FDS) and building models like Building Information Modeling (BIM) to construct a virtual fire sce-

nario by setting parameters for specific buildings. Other studies emphasized the personnel object. Human behavior is complex and affected by many factors. Only studying fire development cannot reduce human casualties well. To solve the problem of considering the complicated individual behavior, many studies focused on the evacuation process in indoor fire scenarios. The common research method is establishing a virtual scenario and simulating evacuation situations using a path search algorithm [19–22]. Combining Virtual Reality (VR), BIM, and Geographical Information System (GIS), many studies have been conducted to improve the existing path search algorithms to raise efficiency, which has become a research hotspot. Among them, VR technology can also be used in the training field to train people to evacuate from fires [23], widening the practical application of VR. Based on the above virtual scenario simulation, to further meet individual evacuee needs, the real-time evacuation system was developed and applied [24,25]. Through mobile phones and other terminal devices, the location of evacuees can be obtained dynamically. Combined with the fire data transmitted by indoor sensors, the path planning can be carried out in real-time, and the results can be returned to the user's mobile phone or displayed on the dynamically changing signs. However, such systems have yet to gain popularity due to cost and other concerns.

### 2.2. Knowledge Graph

#### 2.2.1. General Knowledge Graph

A Knowledge Graph (KG), developed from the Semantic Web, is a structural representation of facts consisting of entities, relations, and semantic descriptions. Entities can be real-world objects and abstract concepts, and relations represent the relations between entities [11]. By extracting knowledge from different sources, KG can process large amounts of data and derive valuable knowledge through various applications. Neo4j, FlockDB, AllegroGraph, and other graph databases can store the extracted entities and relations. To date, KG has been widely used in diverse ways. Based on KG, general applications include knowledge reasoning [26], generating explainable recommendations [27], question answering [28], and information retrieval [29]. Recent advances have focused on knowledge representation learning [30], knowledge graph embedding [31], and knowledge graph completion [32]. With the rapid development of KG technology, it is easier to manage mass data [33,34] and facilitate data conversion to knowledge [35,36].

#### 2.2.2. Disaster Domain Knowledge Graph

KGs in the domain of disaster are significant in intelligent response to various disasters, which deserve further exploration. However, despite the rapid development of KG technology, some shortcomings still need to be further studied and solved. Data quality cannot generally be effectively guaranteed due to noisy data sources like social media. Information interoperability is still not fully realized due to insufficient open data and methods. The dynamic change in knowledge is not well considered because of the lack of temporal properties [37]. Meanwhile, in contrast, the development of general KGs has outpaced the domain KGs, which cannot adapt to professional needs [38]. Therefore, many studies aimed to solve the above shortcomings and constructing domain KGs in recent years. In particular, in disaster prevention and management, higher requirements are put forward for data quality, temporal and spatial attributes, and large data volume [39,40]. Table 1 shows the general differences of KGs.

**Table 1.** Comparison of different KGs.

| | KG Type | Development Degree | Volume Status | Professional Requirement | Dynamic Change Requirement | Spatio-Temporal Property Requirement |
|---|---|---|---|---|---|---|
| | General KG | High | Large | Low | Low | None |
| Domain KG | Disaster Domain | Low | Small | High | High | High |
| | Other Domain | Middle | Large | High | Middle | Low |

As disaster-related data come from various sources with different formats, traditional research methods are challenging to integrate a large number of heterogeneous data. Existing studies cannot effectively reflect the spatio-temporal and semantic relations between disaster components, so conducting in-depth and intelligent analysis of disasters is hard. Therefore, with its powerful data integration ability and rich application ways, KG can provide strong support for the development of disaster research. For each familiar disaster, the corresponding KG was constructed, including earthquakes [41], landslides [42], and epidemics [43], focusing on integrating large-scale heterogeneous data. Many studies combined knowledge reasoning, recommender systems, and other applications to provide technical support for disaster management, emergency response, and disaster simulation. Based on the existing KG applications, considering the spatio-temporal characteristics of disaster components, some studies introduced spatio-temporal attributes into the KG to achieve disaster prediction and other functions [44].

*2.3. Summary*

There are studies on indoor fire emergency evacuation from different perspectives; however, the indoor fire and emergency evacuation process has been analyzed independently [45,46], but few studies have considered the relations among components of indoor fire events. At the same time, evacuation efficiency is affected by various factors. Such as individual characteristics, evacuation route conditions, temperature, smoke concentration, etc. [47,48]. The status of these factors varies with time and space environment, so simple numerical simulation has limitations. KG technology can provide solutions to the limitations and deficiencies mentioned above. To the best of our knowledge, there is currently no knowledge graph for indoor fire emergency evacuation. Hence, the research goal of this study is to construct a knowledge graph to facilitate indoor fire emergency evacuation.

**3. Materials and Methods**

*3.1. Study Area*

Kangleyuan Apartment for Older People was a private nursing home in Lushan, Pingdingshan, Henan, China (Figure 1), established in 2010 and covered an area of 30 acres. The whole nursing home was divided into three levels according to self-care ability, which can accommodate 150 older people. On the evening of 25 May 2015, a short circuit in the wire happened in the building of older people who were mainly disabled or paralyzed and could not care for themselves or even move. The short circuit ignited the surrounding combustibles, and the fire spread to the building. The fire accident in Kangleyuan Apartment for Older People eventually caused 39 deaths and a direct property loss of RMB 20.645 million due to evacuees' low mobility, illegal flammable materials in construction, and inadequate supervision from local authorities. The Henan Pingdingshan '5.25' Fire Accident was a severe fire accident. Government departments were involved in the accident rescue and investigation. Since the accident involved various evacuees, it is a typical indoor fire emergency evacuation case.

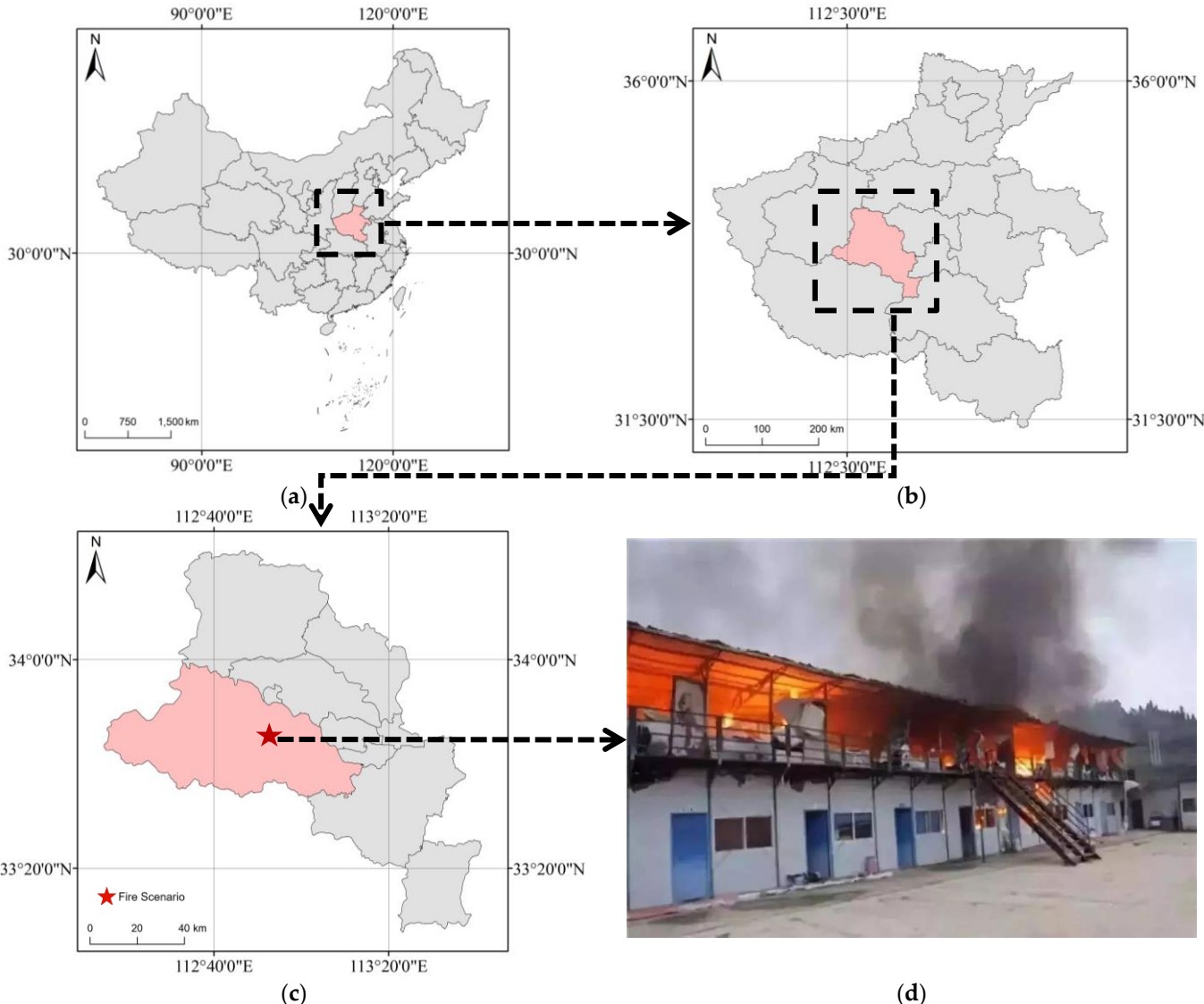

**Figure 1.** Study area. (**a**) Henan's location in China; (**b**) Pingdingshan's location in Henan; (**c**) Lushan's location in Pingdingshan; (**d**) Fire scenario.

### 3.2. Overall Technical Process

The construction of the knowledge graph for indoor fire emergency evacuation is divided into three main steps: ontology construction, knowledge representation model construction, and knowledge acquisition and processing (Figure 2). Since indoor fire emergency evacuation events change constantly, it is crucial to understanding its development process to improve evacuation efficiency. Therefore, we first conducted a spatio-temporal process analysis to determine the basic components of indoor fire emergency evacuation, including fire, building, personnel, time, and space. Second, we constructed the domain ontology consisting of these basic components based on various knowledge sources. To clarify the relations among the components, we carried out the formal representation of the ontology. We proposed a four-tuple representation model and clarified the relations among each component. Then, we collected and organized data from multiple data sources. However, the data in the domain of indoor fire emergency evacuation were mainly unstructured. We located core paragraphs related to fire emergency evacuation through manual reading based on their titles, identified valuable sentences, and manually extracted them to form a corpus. Based on the corpus, we judged word by word whether it is the entity relevant to the emergency evacuation event of indoor fire according to the representation model

above. Then, relations were extracted according to the semantic relation of the sentences. Relation entities were used for connecting all entities to form triples to be stored in the graph databases to obtain a complete knowledge graph, which can support visualization and application.

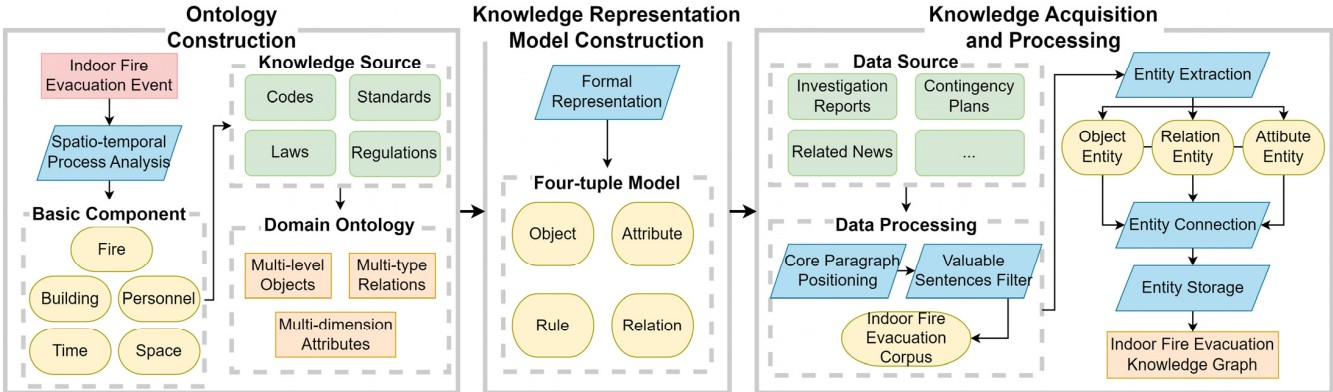

**Figure 2.** Research flowchart of constructing a knowledge graph for indoor fire emergency evacuation.

### 3.3. Ontology Construction

#### 3.3.1. Spatio-Temporal Process Analysis

The evacuation process of indoor personnel in a fire scenario is relatively complex. It is not a single-moving process but a complex process completed under the joint action of individual factors such as age, gender, education level, and environmental factors such as fire products. Indoor fire emergency evacuation can be divided into three stages: pre-movement, movement, and post-movement (Figure 3).

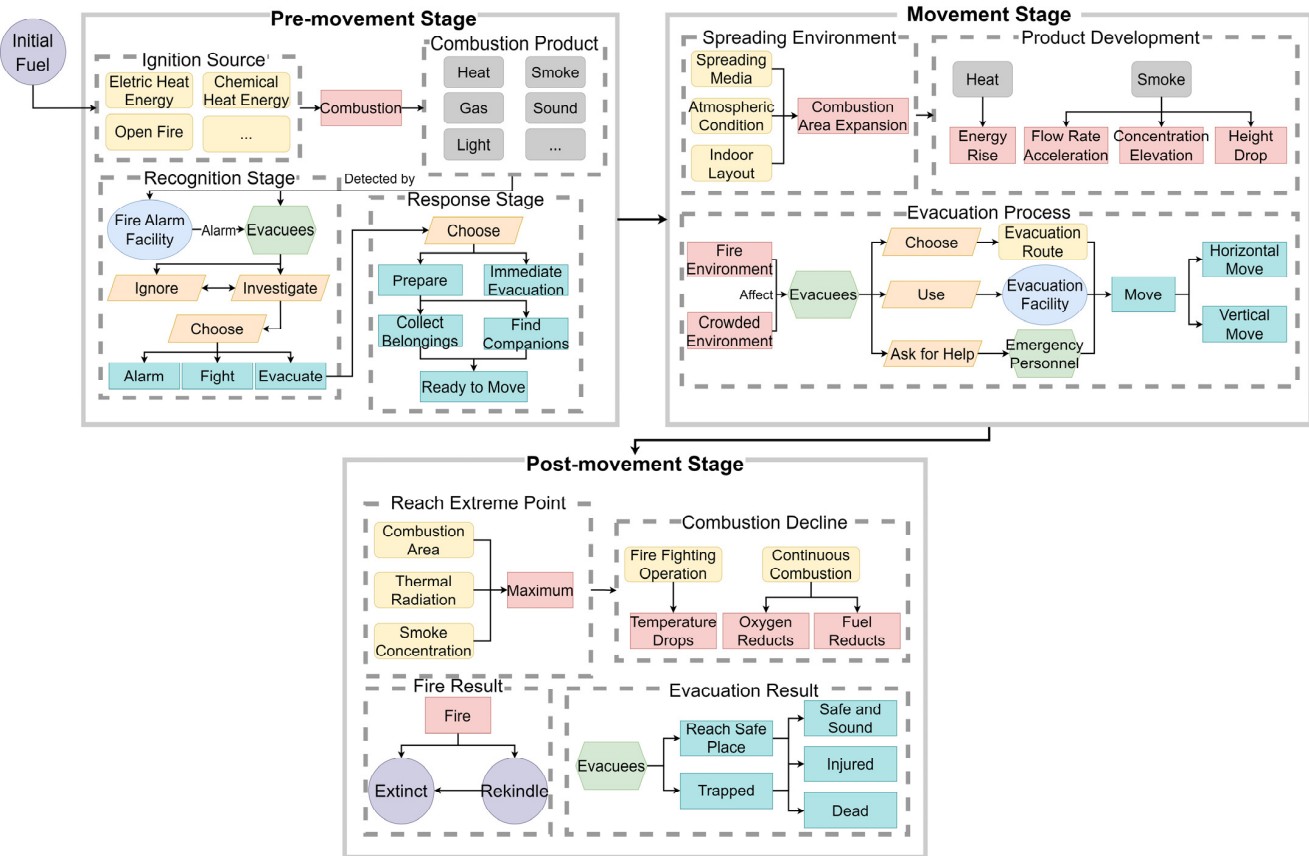

**Figure 3.** The development stages of indoor fire emergency evacuation.

In the pre-movement stage, the fire is in its initial stage. The initial fuel starts to burn under the action of ignition sources such as electric heating energy, chemical heat energy, and so on, producing heat, smoke, and other fire products. When the fire products attract attention, the first sub-stage of the pre-movement stage starts the recognition stage. At recognition stage, the main event is the recognition of the fire by the evacuees. Evacuees can cognize the fire products themselves or be informed by the fire alarm facilities. After obtaining the corresponding information, indoor personnel may have two choices. One choice for indoor personnel is to ignore the information obtained and stay where they are. The other is to conduct investigations, such as observing independently or asking others. Due to the differences and changes in the information obtained, ignoring and investigating are interchangeable. After the fire is confirmed, the evacuees need to make another round of choices, including alarming, trying to extinguish the fire, or evacuating. The alarm includes calling the firefighters and alerting other people indoors. If the fire is controllable, some personnel take the initiative to extinguish the fire. If the fire is out of control, or the indoor personnel do not know how to fight fire, the indoor personnel choose to evacuate.

Making these decisions is the goal of the recognition stage. The indoor personnel who decide to evacuate will move on to the next sub-stage, the response stage. The evacuation movement has not yet begun, and options are still being made. The first choice is whether to prepare for evacuation first or evacuate immediately. The preparation before evacuation includes sorting out and collecting personal belongings and finding companions. After that, indoor personnel are ready for action. The choices in the response stage are based on the personnel's recognition of the fire situation. If one thinks the fire is already severe, one will be more inclined to evacuate immediately.

The movement stage is the main component of the evacuation process. The fire is also developing when most people indoors are ready to evacuate. The combustion area begins to expand based on a spreading medium, good ventilation conditions, an open indoor layout, and other conditions. At the same time, the continuous addition of new fuels makes the rapid development of combustion products. It includes the rise of thermal energy, the acceleration of flow rate, the increase in concentration, and the decrease in the height of smoke. Under the combined influence of a fire environment and crowded people, evacuees must choose evacuation routes, use evacuation facilities, and seek help from emergency personnel if necessary. Then, evacuees will move to safety, which requires the horizontal movement of the same floor and the vertical movement of the cross-floor. These movements have different characteristics and influencing factors.

The post-movement stage includes the evacuation result after the main part of the evacuation. With fire parameters such as combustion area, thermal radiation, smoke concentration, etc., reaching an extreme value, the fire begins to turn to the recession stage. The firefighting action makes the temperature drop, and the continuous burning consumes oxygen and fuels. The fire may rekindle under certain conditions but eventually becomes extinct. And evacuees also come to the end of the evacuation. Some evacuees reach a safe place, and some are trapped indoors, waiting for rescue. There are three possible outcomes for evacuees: safe and sound, injured, and dead. Due to the dangers in the evacuation process, even if the evacuee has reached a safe place, the evacuee may eventually die. But, trapped people may be safe and sound if the trapped area is relatively safe and rescued in time. Therefore, the outcome of evacuation varies enormously from individual to individual.

### 3.3.2. Domain Ontology

Based on the analysis in Section 3.3.1, we widely obtained knowledge from relevant knowledge sources, including codes, standards, laws, and regulations from the USA, China, and other countries or regions worldwide (Table 2).

**Table 2.** Knowledge sources of indoor fire domain ontology.

| Knowledge Type | Knowledge Source | URL |
|---|---|---|
| Codes and Standards | National Fire Protection Association Code (USA) | https://www.nfpa.org/Codes-and-Standards/All-Codes-and-Standards/List-of-Codes-and-Standards (accessed on 20 August 2023) |
| | Fire Protection Vocabulary (China) | https://openstd.samr.gov.cn/bzgk/gb/std_list?p.p1=0&p.p90=circulation_date&p.p91=desc&p.p2=GB/T%205907.1-2014 (accessed on 20 August 2023) |
| | Code of Practice for Fire Safety in Buildings (Hong Kong) | https://www.bd.gov.hk/en/resources/codes-and-references/codes-and-design-manuals/fs2011.html (accessed on 20 August 2023) |
| | Code of Practice for Fire Precautions in Buildings (Singapore) | https://www.scdf.gov.sg/firecode/table-of-content (accessed on 20 August 2023) |
| Laws and Regulations | Fire and Security National Regulations (Europe) | https://cfpa-e.eu/national-regulations/ (accessed on 20 August 2023) |
| | Fire Protection Law (China) | http://www.cfpa.cn/home/Specialnews/show.html?specialnews_id=121 (accessed on 20 August 2023) |
| | Fire Safety Regulations (India) | https://firesafetysecurityindia.com/fire-safety-regulations-in-india/ (accessed on 20 August 2023) |

(1)   Object–Attribute

Object and attribute concepts are formed by manual summarization (Appendix A, Table A1). The main objects of indoor fire emergency evacuation are classified into levels (Figure 4). The attribute knowledge with detailed granularity, i.e., the important parameters describing the whole process of fire and evacuation, as well as the more systematic division of indoor fire development stages and emergency evacuation stages. These attributes are provided in the fire investigation reports and related simulation studies, thus serving as the attribute nodes of the knowledge graph.

Among them, the addition of time and space objects makes the description of the emergency evacuation process of indoor fire more specific. For time objects, we chose the ignition time, the detecting time, the alarm time, etc. The duration can be calculated based on time points. I2D Duration means the time from the start of the fire to the time of its detection by the personnel. Because of the rapid development of the fire, small differences in the timing of the evacuation have a greater impact than expected. At the same time, time objects can also relate to the state development of evacuation options. For example, Exit A can pass when the fire starts, but it is blocked by smoke 5 min later. Therefore, time objects can also reflect dynamic changes. The influence of space objects on personnel evacuation is also significant. The location of the fire, the initial location of the evacuees, the location of evacuation options, etc., are beneficial in analyzing evacuation behavior.

Some attributes have the name of 'State'. The 'State' attributes are dynamic and complex. Specific examples of the 'State' attributes are summarized based on relevant survey reports and research cases (Appendix A, Table A2).

(2)   Relation

The relations are used to represent inter-object relations and inter-object-attribute relations. In this study, relations are divided into three main types: inter-object relation, object-attribute relation, and inter-attribute relation. And inter-object relations are further divided into five types, including whole–part relations, parent-child inheritance relations, action class relations, temporal relations, and spatial relations. Some examples of relations are shown in Appendix A, Table A3.

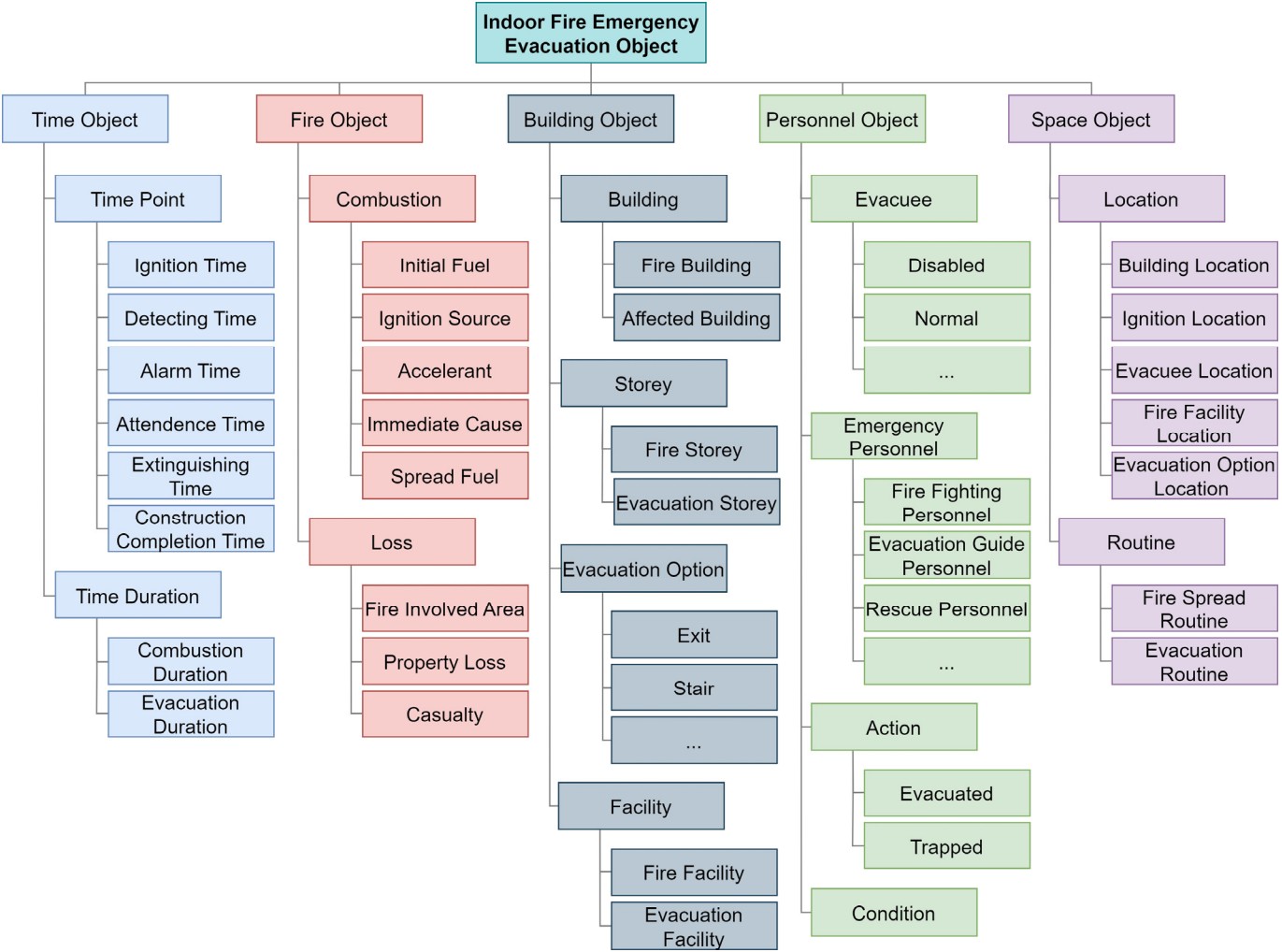

**Figure 4.** Classification of objects of indoor fire emergency evacuation.

*3.4. Formal Representation*

To clarify the relations among the components, we carried out the formal representation of the ontology. We proposed a four-tuple representation model and clarified the relations in detail. Knowledge representation is the premise and foundation for knowledge organization and knowledge application of indoor fire emergency evacuation. We divided the event into four components: object, attribute, relation, and rule, which can be represented formally as follows:

$$IFEO = <IFEO\text{-}Objects, IFEO\text{-}Attributes, IFEO\text{-}Relations, IFEO\text{-}Rules>. \tag{1}$$

In the knowledge representation model, the IFEO-Objects describe the composition of objects of the indoor fire emergency evacuation event. The IFEO-Attributes define the static attributes of the object itself and the dynamic attributes in different states. The IFEO-Relations specifies the main types of relations, such as inter-object, object-attribute, and other types of relations in the above-mentioned components of the knowledge representation model. The IFEO-Rules represent the range of values, types, and constraints of combinations to support applications. With the above four as the main body, the knowledge expression model is constructed to systematically represent indoor fire emergency evacuation events.

(1)    Object

The IFEO-Objects describe the constituent objects of an indoor fire emergency evacuation event. After analysis of the spatio-temporal process, according to the characteristics of indoor fire emergency evacuation events and fields involved, the IFEO-Objects can be split into five categories: fire objects, building objects, personnel objects, time objects, and space objects, and each category of objects can be further subdivided.

We let O denote an object and n be the number of object levels, so IFEO-Objects can be formally defined as follows:

$$\text{IFEO-Objects} = <O_1, O_2, O_3, \ldots, O_n>. \tag{2}$$

(2)    Attribute

The IFEO-Attributes records the detailed information of object entities and is an important basis for knowledge queries. In this study, the attributes of indoor fire emergency evacuation are divided into two categories: Static attributes S and Dynamic attributes D. Attribute S is used to characterize the static attributes of entities, such as building basic information that is inherent to the class of objects and does not change with the development of events, and such attributes are used to distinguish things; Attribute D is used to describe the attributes of entities that change dynamically with the evolution of time and space in different states of fire development and evacuation.

We denote the object by C, m and n denote the number of attributes of the two categories. Therefore, the attribute can be represented as follows:

$$\text{IFEO-Attributes} = <C, S_1, S_2, S_3, \ldots, S_m, D_1, D_2, D_3, \ldots, D_n>. \tag{3}$$

(3)    Relation

Certain connections between objects and attributes constitute indoor fire emergency evacuation events. These connections make the discrete objects and attribute nodes connected into a whole, thus facilitating relevant queries in specific applications. The connections between objects and attributes form the IFEO-Relations of the representation model. In this study, we classify relations into three types: inter-object relation, object-attribute relation, and inter-attribute relation. Among them, inter-object relations can also be subdivided into five types. Each type is specifically reflected in the IFEO-Relations.

We let A and B be two objects or two attributes or one object and one attribute; $R_{AB}$ denotes the relation that exists between A and B. Then, the relation can be represented as follows:

$$\text{IFEO-Relations} = <A, R_{AB}, B>. \tag{4}$$

(4)    Rule

Rule is the constraint of the above object, attribute, and relation in the form of value type and range. A can represents one kind of object, attribute or relation. And R represents the rule. So, the rule can be represented as follows:

$$\text{IFEO-Rules} = <A, R>. \tag{5}$$

*3.5. Knowledge Acquisition and Processing*

3.5.1. Data Source and Processing

The data sources of the indoor fire emergency evacuation knowledge graph are shown in Table 3, mainly including the investigation reports from the Ministry of Emergency Management of China, the news reports from CCTV, the Huanqiu network, and other information sources, which are primarily unstructured data. Table 3 presents the following information: data type, data source, data volume, data proportion, and data description. The volume of data is measured by the number of valuable sentences (NVS). In addition, investigation reports may involve many fields, including responsibility identification and

punishment, etc., which are inconsistent with the focus of this study. The same situation also exists in news reports and other sources. Considering the difficulties above, we first conducted core paragraph positioning to roughly filter paragraphs for possible entities based on the general meaning. Based on these paragraphs, we further selected sentences containing entities according to the narrative characteristics of different information sources. With these potentially valuable sentences stored, a corpus of indoor dire evacuation was constructed for further extraction.

**Table 3.** Data Introduction.

| Data Type | Data Source | Data Source | Data Volume (NVS) | Data Proportion | Data Description |
|---|---|---|---|---|---|
| Semi-structured text | Search Engine | Baidu | 325 | 15.33% | The search results with the fire event names as the keywords, containing basic information |
| Unstructured text | Investigation Report | Ministry of Emergency Management | 642 | 30.28% | The official investigation reports, covering the fire development and evacuation details |
| | News Report | CCTV | 244 | 11.51% | Authoritative news media, containing voice-over text collated from news videos, and text press releases |
| | | Huanqiu Net | 203 | 9.58% | International news, relatively detailed fire information, with specific introduction and related photos |
| | | Souhu Net | 107 | 5.05% | Reprint of other news media, including text introduction and interview content, playing a complementary role |
| | Social Media | WeChat | 301 | 14.20% | Introduction articles from WeChat Official Accounts, edited and organized text |
| | | Weibo | 298 | 14.06% | Short text, official and personal; the user scale is large |

### 3.5.2. Entity Extraction

Based on the knowledge representation model and indoor fire emergency evacuation corpus, we extracted the corresponding entities, including object and attribute entities. Part of the results is shown in Table 4. Since there could be fire buildings and affected buildings with multiple floors, the situation of evacuees and facilities in each building and on each floor may be different. Therefore, when extracting entities, we considered each building and floor in detail, using labels with prefixes to clarify the affiliation of entities. In addition, extracted attributes are stored in numerical form, and attributes of the same type share a uniform unit. For example, the unit of area is square meter, the unit of loss is yuan, and the unit of time is minute.

**Table 4.** Part of the extracted entities.

| Label | Content |
|---|---|
| Fire Building_Number of Storeys | 1 |
| Fire Building_Area Covered | 745.8 |
| Fire Building_Structure | Steel Structure |
| Fire Building_Purpose | Residential Building |
| Evacuation Storey | First Floor |
| First Floor_Number of Occupants | 56 |
| First Floor_Evacuation Option | East Exit |
| East Exit | Unblocked |
| Accelerant | Oxygen |
| Spread Fuel | Polystyrene Material |
| Fire Involved Area | Whole Building |
| Fire Involved Area_Area | 745.8 |
| Property Loss_Amount | 2064.5 |
| Casualty_Death Toll | 39 |
| Casualty_Number of Injured | 6 |
| Occupant_Disabled_Proportion | 93% |
| Ignition Time | 25 May 2015 19:30 |
| Combustion Duration | 0:50:00 |

### 3.5.3. Entity Connection

Relation entities (Table 5) are used for connecting all entities to form triples so as to be further stored in graph databases. Entity connection is of importance for the knowledge graph construction. Different types of relations give entities rich semantic, temporal, and spatial connections, which play an important role in the future query service.

**Table 5.** Examples of entity connection.

| Node | Relation | Node |
|---|---|---|
| Kangleyuan Apartment for Older People | Contains | Facility |
| Facility | Contains | Barrier-free Evacuation Facilities |
| Barrier-free Evacuation Facilities | Contains | Barrier-free Evacuation Elevator |
| Kangleyuan Apartment for Older People | Area Covered | 745.8 |
| Kangleyuan Apartment for Older People | Structure | Steel Structure |
| Kangleyuan Apartment for Older People | Purpose | Residential Building |

### 3.5.4. Entity Storage

When entity extraction and entity connection are completed, constructing a knowledge graph of indoor fire emergency evacuation comes to the last step: knowledge storage. In this study, we selected the well-known graph database Neo4j to store the connected entities in the database in triples (Figure 5). Neo4j is one of the best choices when choosing a graph database with the most important features, such as a flexible schema and a powerful SQL-like query language called Cypher [49]. Neo4j also supports massive data relational operations [50], so we can efficiently query and update graphs in Neo4j. Massive data can be stored and queried for indoor fire emergency evacuation, allowing a more intuitive visualization of knowledge and facilitating users to understand the indoor fire emergency evacuation field fully. In addition, the query statements provided by Neo4j facilitate the application of the knowledge graph.

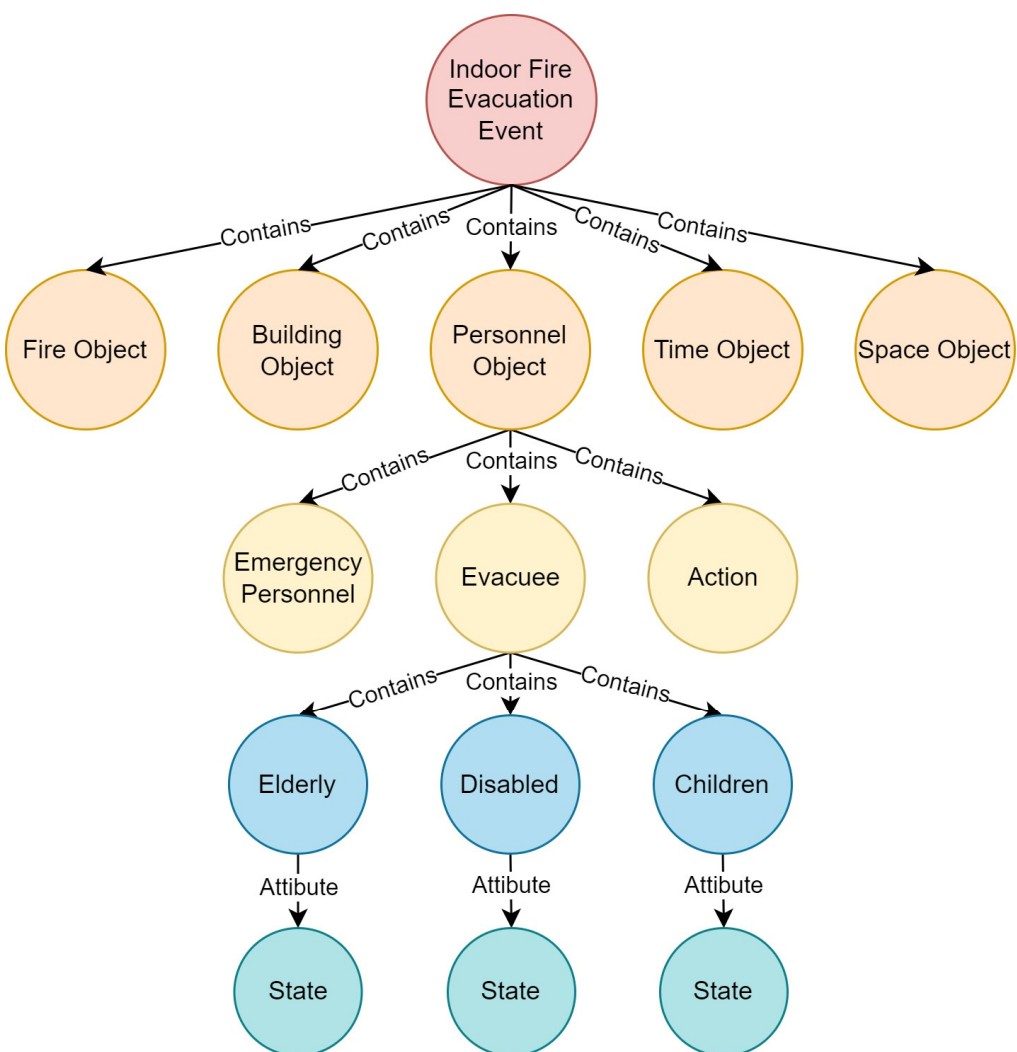

**Figure 5.** An illustration of knowledge storage of indoor fire emergency evacuation.

## 4. Results

### 4.1. Knowledge Graph for Indoor Fire Emergency Evacuation

In this study, we collected and organized 25 representative indoor fire events in China and carried out entity extraction, entity connection, and entity storage operations according to the methods mentioned in Section 3. A total of 1852 nodes and 2364 relations were obtained and connected as triples to be stored in Neo4j (Figure 6). The construction process of the knowledge graph for indoor fire emergency evacuation is carried out through the

conversion of the knowledge representation model into corresponding components. The objects and attributes are converted into nodes, and the relations are converted into edges connecting each node in the graph. Together, the nodes and edges constitute the complete knowledge graph. In the knowledge graph of indoor fire emergency evacuation, various objects and their attributes in the case of fire incidents are stored in detail, connected by relations. Among them, the fire objects store the whole process of fire development and the damage caused by the fire. The process of fire development includes the combustion elements of a fire, such as the source of ignition, initial fuel, and spreading fuel. Damage caused by fire includes property damage and personnel casualties. In addition, the time and space objects classify spatio-temporal information according to different dimensions and establish spatio-temporal relations.

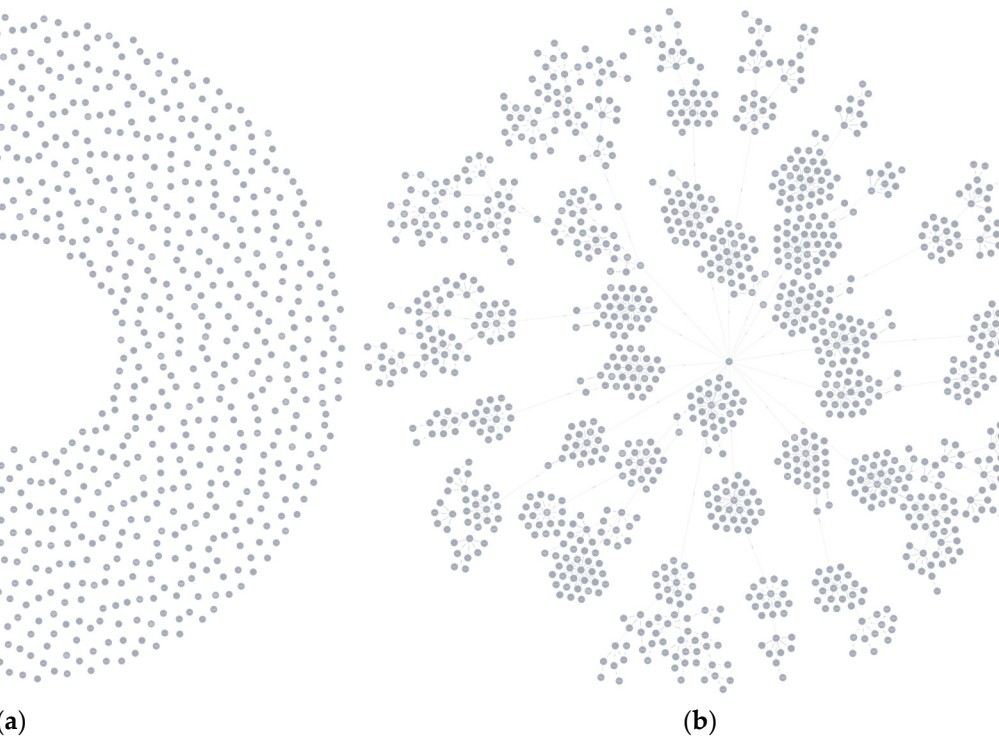

(**a**)                                                          (**b**)

**Figure 6.** Knowledge graph for indoor fire emergency evacuation. (**a**) All nodes; (**b**) Nodes with relations.

### 4.2. Case Study

According to the study area selected in Section 3.1, we focused on the Kangleyuan Apartment for Older People in Henan, China. Neo4j provided a convenient query function, and we only needed to query the name of the fire event. Then, we could quickly locate all the object entities related to the event. Figure 7 shows some of the query results in Neo4j Desktop for its knowledge content. We used different colors to represent different types of nodes, including object and attribute nodes, and we also used different colors to distinguish objects of different levels. By means of color grading, we could observe the hierarchy of the event clearly. Different objects, different levels of objects, and their relations were organized uniformly, which provided strong support for further application according to the graph.

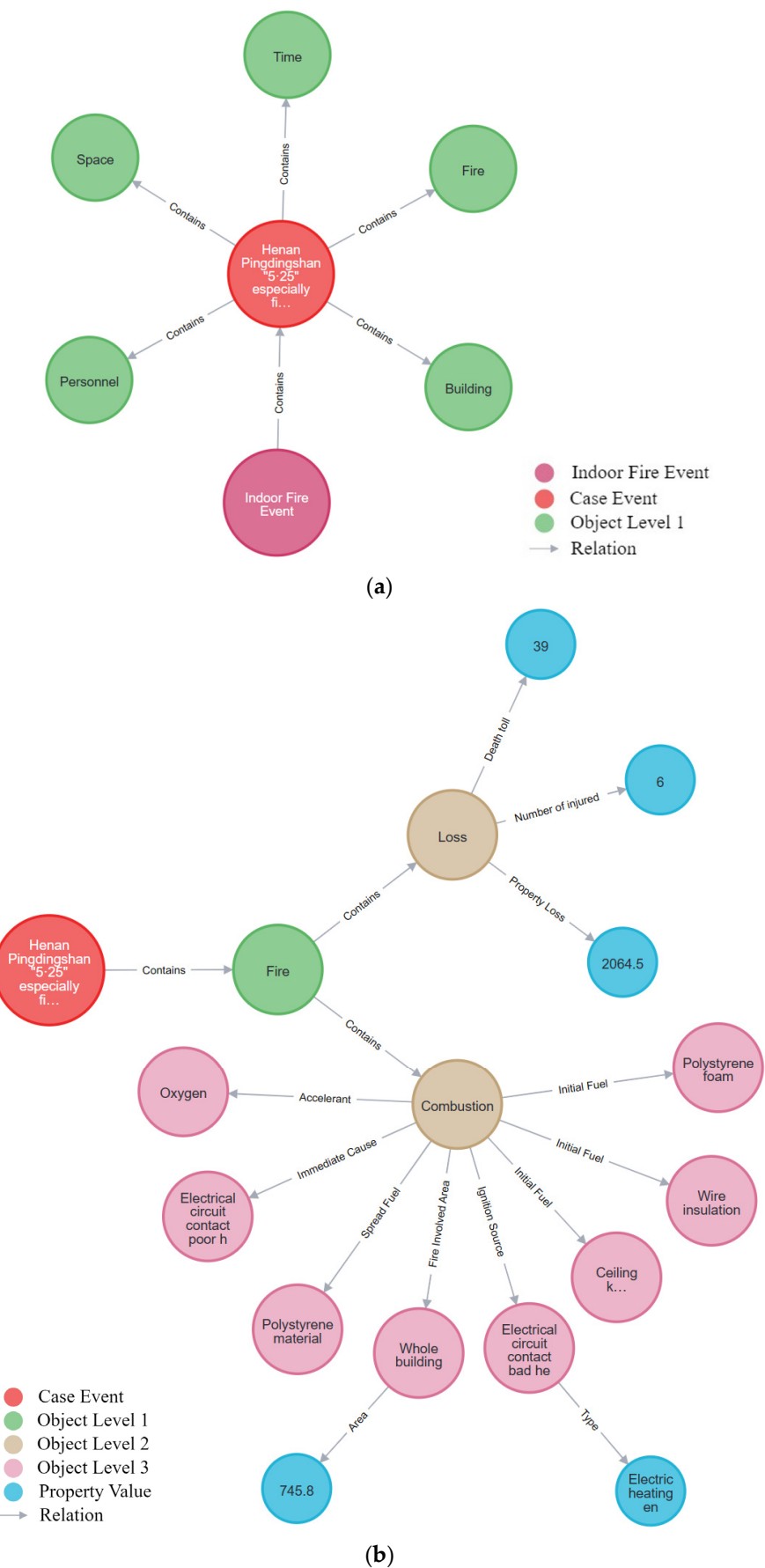

**Figure 7.** *Cont.*

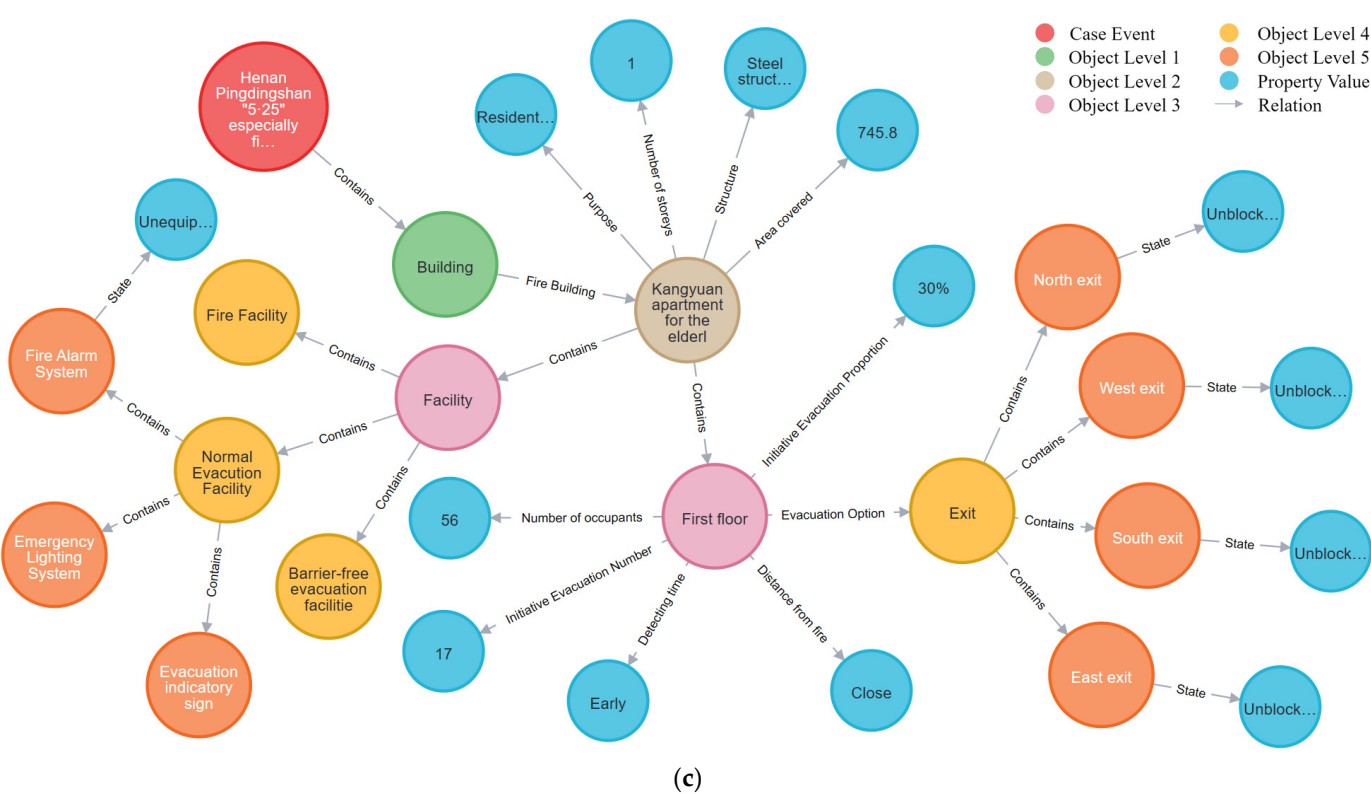

**(c)**

**Figure 7.** Part of the query results in Neo4j. (**a**) Basic structure of the knowledge graph; (**b**) Part of fire objects; (**c**) Part of building objects.

### 4.3. Analysis

We first designed a quantitative evaluation method to test the above construction's accuracy. We conducted random sampling statistics on the accuracy of different nodes and relation extraction. We evaluated the extracted nodes using the accuracy rate, calculated as shown in (6), where $N_{right}$ is the number of correct nodes, and $N$ is the number of all nodes. The results are shown in Table 6, proving that the accuracy of manual extraction is very high. Then, we designed several application methods for the proposed knowledge graph.

$$Accuracy = \frac{N_{right}}{N}. \tag{6}$$

**Table 6.** Accuracy of node and relation extraction (random sampling).

| Type | N | $N_{right}$ | Accuracy |
|:---:|:---:|:---:|:---:|
| Fire Object Nodes | 100 | 100 | 100% |
| Building Object Nodes | 120 | 115 | 95.83% |
| Personnel Object Nodes | 150 | 144 | 96% |
| Time Object Nodes | 30 | 29 | 96.67% |
| Space Object Nodes | 40 | 37 | 92.50% |
| Relations | 500 | 482 | 96.40% |

The application of the knowledge graph for indoor fire emergency evacuation is query service, which mainly consists of three types: (1) query the detailed information of event objects; (2) query the results of personnel evacuation under specific conditions; (3) query the events that have spatial and temporal similarities.

Neo4j provides a quick node and relation query function. To fully use the query function, the labels or names of nodes were named with object, attribute, and relation names when importing nodes and relations in bulk. Therefore, after importing instance data, the knowledge graph could quickly find specific objects and relation entities using labels and other information.

For the first type of query, for the Henan Pingdingshan '5.25' Fire Accident in China, the fire combustion information could be queried through the starting node of the relation to all the related attribute entities. Taking 'Combustion' as the root node, we could query all the sub-level nodes and relevant attribute values and then query the combustion information of the Henan Pingdingshan '5.25' Fire Accident in China (Figure 8). The query result showed that the initial combustible materials were flammable and combustible fuels such as wire insulation, polystyrene foam, and ceiling wood keel. Meanwhile, the overfire area was the whole building, which covered 745.8 m². In addition to the fire combustion information, the fire building and evacuee information could be queried similarly.

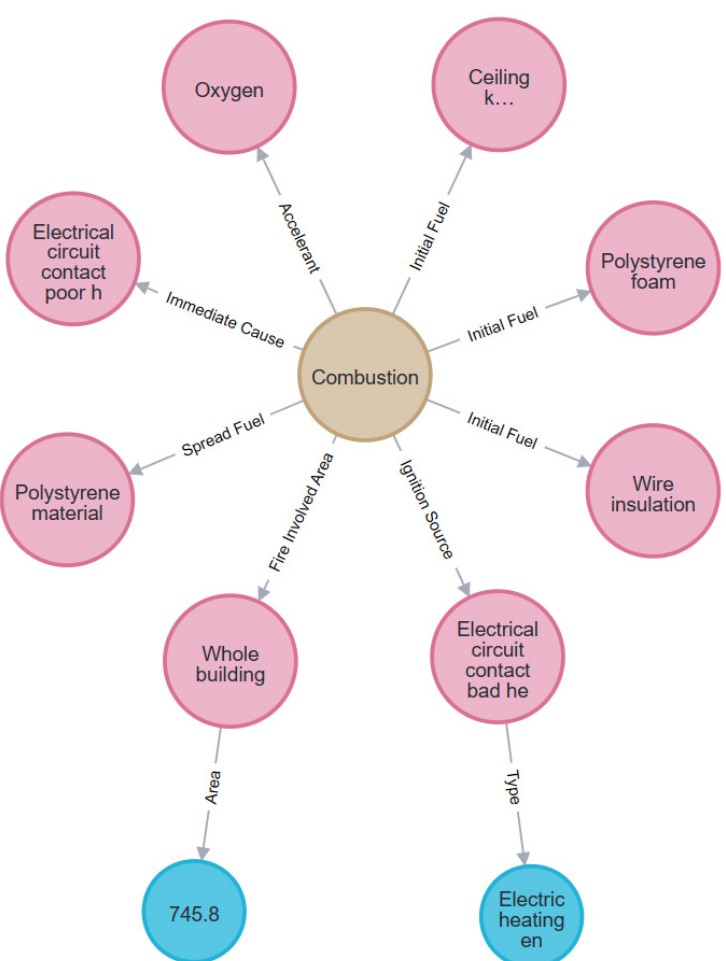

**Figure 8.** Query results of combustion information.

For the second type of query, querying the evacuation results of people under specific conditions could be further refined into two sub-problems (1) single impact condition; (2) multiple impact conditions. Since the evacuation of disabled persons is less of a concern during the emergency evacuation of indoor fires, disabled persons are often more challenging to evacuate than normally mobile persons, thus causing enormous casualties. Therefore, the query with disabled people as the main subjects could further understand the current status of the evacuation of disabled people and provide suggestions and a theoretical basis for designing indoor places for disabled people.

For the first problem, the conditions of evacuation guide personnel in the personnel object and evacuee locations in the spatial object were tested in Neo4j with disabled persons as the main subjects. The success rate of evacuating people with disabilities is significantly higher when evacuation guide personnel guide them, according to relevant research literature and news reports. In addition, being on a higher floor at the time of the fire or other locations away from the exit can lead to the failure to evacuate people with disabilities. The query results showed that the' Evacuated' node was returned under the condition that the evacuation guide personnel provide guidance (Figure 9a). The result indicated that the evacuation guide personnel play an essential role in the evacuation of disabled persons. Meanwhile, under the condition that the evacuees are located near the exit, the 'Evacuated' node was also returned (Figure 9b). The result indicated that the spatial location of disabled persons is significant to the successful evacuation. The query results remained consistent with the results summarized in the literature. Besides disabled personnel, the abovementioned conditions influence whether the normal-mobility personnel can be evacuated successfully to a certain extent.

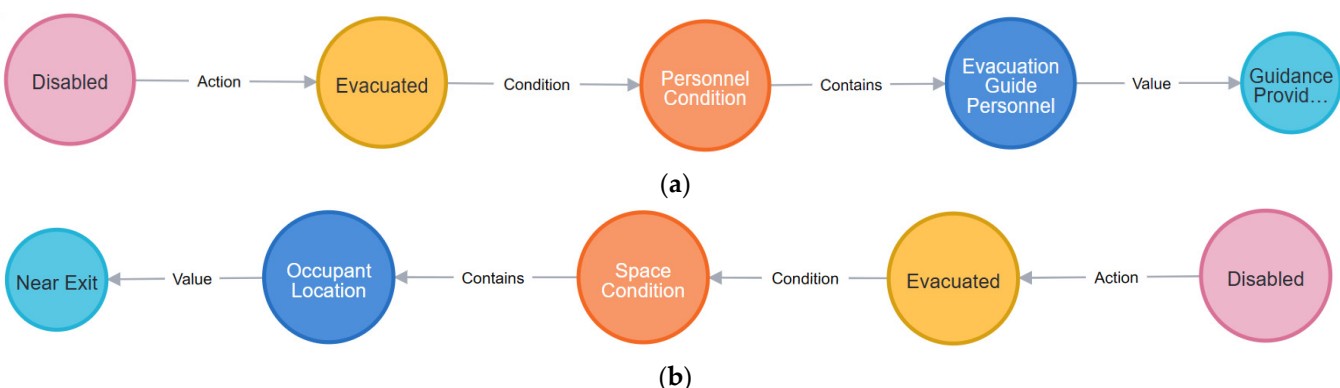

**Figure 9.** Single condition for disabled people to evacuate. (**a**) Personnel condition; (**b**) Space condition.

For the second problem, multiple impact conditions were taken into account. The influence of building and personnel conditions was focused on the query in Neo4j. The presence of barrier-free evacuation indoor facilities specifically designed for various types of people with disabilities played a crucial role in successfully evacuating disabled people in large and complex buildings. We investigated the conditions under which barrier-free evacuation doors, evacuation elevators, and evacuation ramps were unequipped. Also, the rescue personnel and evacuation guide personnel were absent to provide help. Then the 'Trapped' node was returned (Figure 10). The result of the 'Trapped' node illustrated that the absence of major barrier-free evacuation facilities on the premises can seriously affect the evacuation success rate of persons with disabilities as much as the absence of assistance from the rest of the personnel, which is consistent with the expected results.

Finally, the third type of query was based on space and time objects. We input time and space conditions in Neo4j and return event nodes to find indoor fire events with spatio-temporal similarity. For example, we need to know which of the following indoor fire events share similar spatio-temporal characteristics: the I2D (Ignition to Detection) duration is less than 20 min, the D2A (Detection to Alarm) duration is less than 5 min, the ignition location is far from exit, and the occupant location is near exit. We need to input these conditions in Neo4j, and then indoor fire events that meet the above conditions all return (Figure 11). The query application for spatio-temporal similarity helps us to obtain indoor fire events with spatial and temporal similarities, which helps us to compare and analyze multiple events and summarize valuable rules.

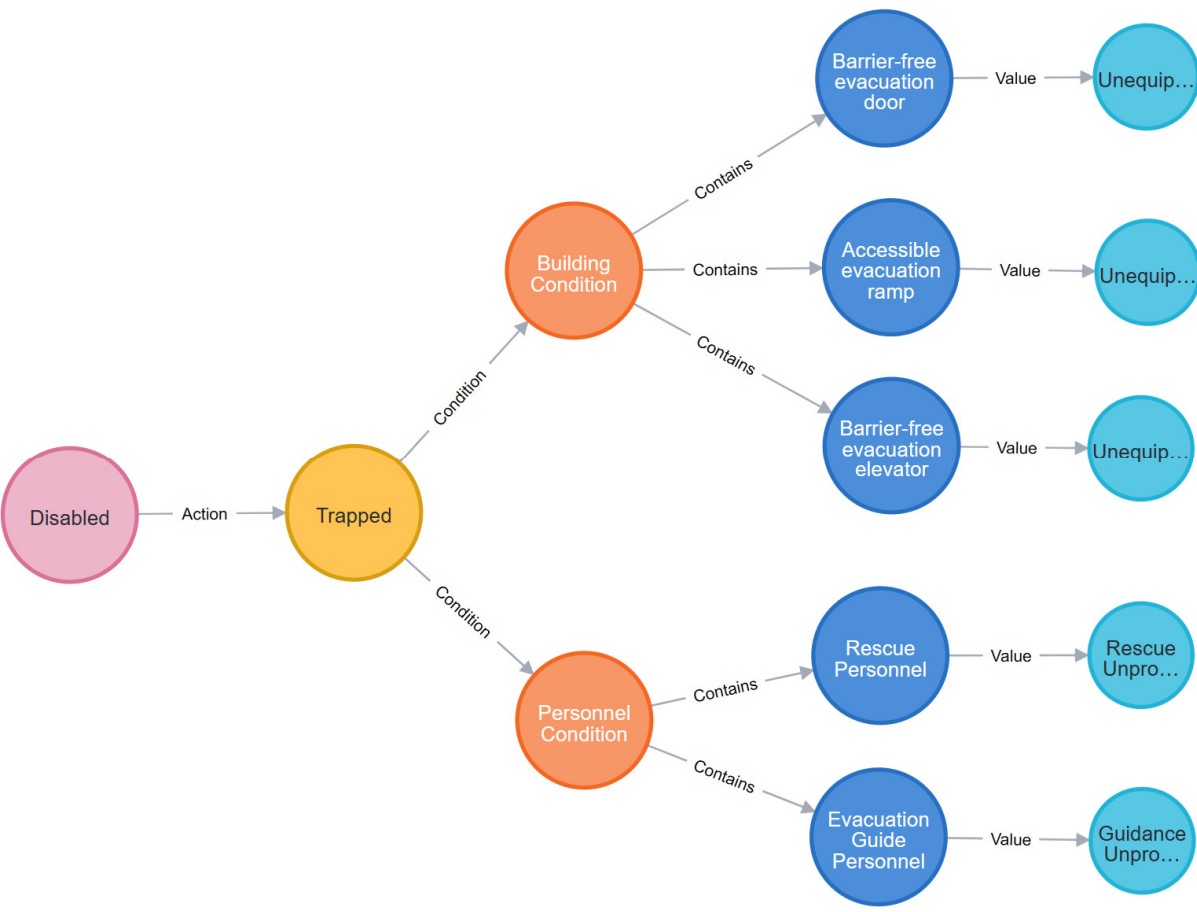

**Figure 10.** Multiple conditions for disabled people to be trapped.

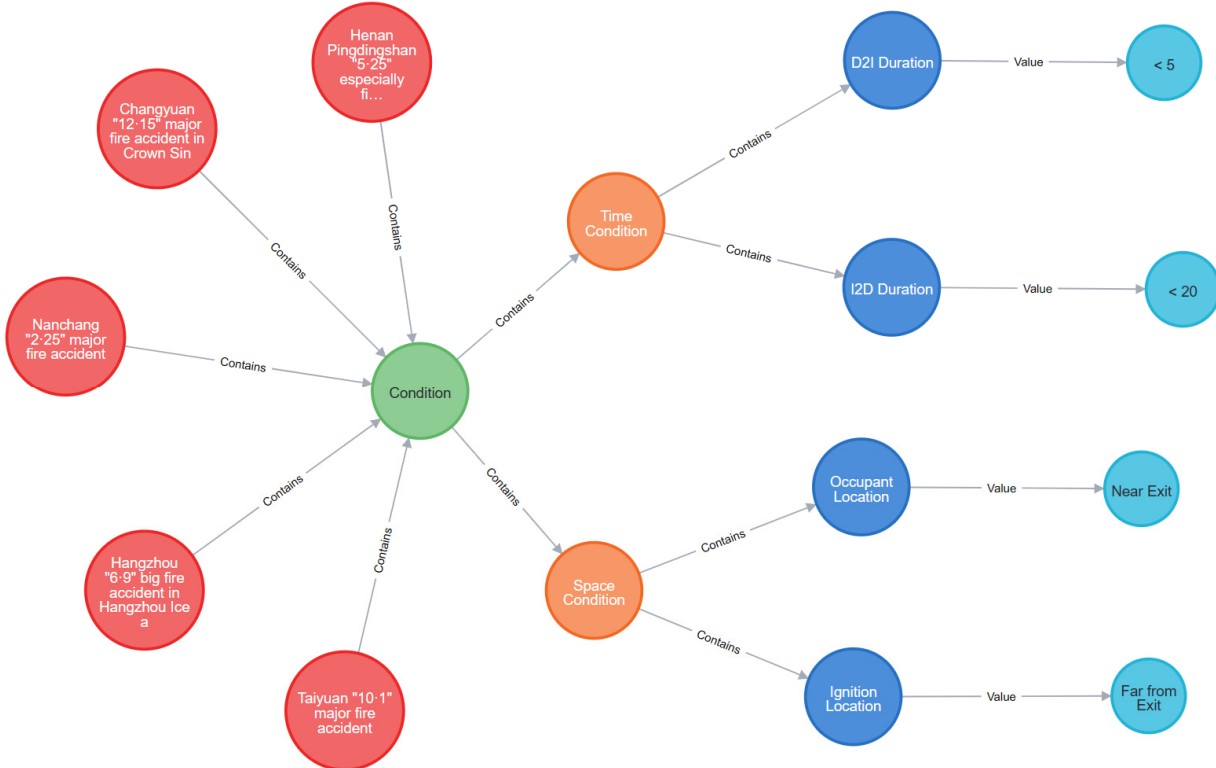

**Figure 11.** Spatio-temporal similarity query result.

## 5. Discussion

The indoor fire emergency evacuation process is a composite event of multiple objects such as fire, building, and personnel organically combined under time and spatial conditions. By constructing the indoor fire domain ontology and designing a knowledge representation model, a knowledge graph for indoor fire emergency evacuation is constructed. Traditional studies in the domain of indoor fire emergency evacuation focused on the separation of fire and evacuation or relied entirely on computer simulation models for numerical simulations. Existing knowledge graphs in the field of indoor fire emergency evacuation fail to focus on evacuees or the evacuation process and cannot achieve the purpose of providing decision support for evacuation. Therefore, a domain ontology is constructed based on a profound analysis of the spatio-temporal process of indoor fire emergency evacuation. And a knowledge representation model is designed at four-tuple: object, attribute, relation, and rule. The proposed knowledge representation model effectively expresses the relations among the objects and forms an organic and unified whole, which is easy to query and apply.

Emergency evacuation of indoor fires does not focus only on the development of the fire. Elements such as the building structure and fire facilities all significantly impact the evacuation process. While recording historical event details from multiple dimensions, conditional analysis of evacuation results is added, thus adding a new application to the query function of the knowledge graph. The impact of a single condition or multiple conditions on evacuation results can be queried through the nodes of the knowledge graph. Due to the classification of evacuees, the query service can focus on special groups, such as people with disabilities. Disabled people are at a significant disadvantage compared to normal mobile people during evacuation. Although the existing building design codes and fire emergency evacuation plans mention the evacuation of disabled people, the codes and plans do not form clear and specific rules to guide the evacuation process of disabled people. Therefore, the knowledge graph constructed in this study has a great prospect of application in evacuation path selection, emergency personnel dispatching, and emergency material deployment. We can support various groups of people, including disabled people, in indoor fire emergency evacuation scenarios. In addition, detailed information queries and spatio-temporal similarity queries also enable users to have a deeper understanding of indoor fire emergency evacuation.

## 6. Conclusions

This study considers indoor fire emergency evacuation the main research object and proposes a four-tuple knowledge representation model based on spatio-temporal process analysis and domain ontology construction. The model systematically describes indoor fire emergency evacuation events in a multi-dimensional manner, which is converted into nodes and relations in the knowledge graph and stored in the graph database Neo4j. The Henan Pingdingshan '5.25' Fire Accident in China, is taken as a case study. In this study, we obtain relevant data and conduct data processing, verifying the knowledge graph's practical value for indoor fire emergency evacuation. The knowledge representation of the study case is specific and detailed under the framework of the knowledge representation model, which can describe indoor fire emergency evacuation events wholly and comprehensively. It can be fast and effective in querying specific information. The research and analysis of the evacuation process of special groups, such as disabled people, are added to improve the breadth of the query function and provide new ideas for the evacuation of disabled people under indoor fire conditions. At the same time, the spatio-temporal similarity query service makes full use of time and space objects, which leads to a deeper understanding of indoor fire emergency evacuation. The results prove that the proposed knowledge representation model and the knowledge graph construction method for indoor fire emergency evacuation are fully feasible and fruitful.

However, this study still has some limitations. First, the events of the knowledge graph we constructed all happened in China. Although the most representative cases are selected

according to the characteristics of the data, there is still a shortage in the number and international applicability. This study is a preliminary attempt to construct a knowledge graph for indoor fire emergency evacuation. Manual extraction methods are used to extract entities and relations. Although the accuracy is high, the efficiency is low. Faced with increasing data in future research, we should use automatic methods such as natural language processing to improve extraction efficiency. Due to the limitations of our data sources, some important parameters, such as fire products, are not effectively extracted. In future research, we should design more valuable application methods, including knowledge reasoning, recommendation systems, etc. More suitable data sources need to be added to extract more detailed information about fires, thereby improving the comprehensiveness of the knowledge graph. Based on the efficient management of massive heterogeneous data, the advantage of the knowledge graph will be maximized to provide decision support for indoor fire emergency evacuation.

**Author Contributions:** Conceptualization, Mingkang Da and Teng Zhong; methodology, Teng Zhong and Mingkang Da; experiments, Mingkang Da; data curation, Mingkang Da; writing—original draft preparation, Mingkang Da and Teng Zhong; writing—review and editing, Teng Zhong and Jiaqi Huang; funding acquisition, Teng Zhong. All authors have read and agreed to the published version of the manuscript.

**Funding:** This research was funded by the National Key R&D Program of China (2021YFE0112300), the Natural Science Foundation of the Jiangsu Higher Education Institutions of China (22KJB170003), and the National Natural Science Foundation of China (42271443).

**Data Availability Statement:** Not applicable.

**Acknowledgments:** We would like to thank the editors and anonymous referees for their constructive suggestions and comments that helped improve this paper's quality.

**Conflicts of Interest:** The authors declare no conflict of interest.

## Appendix A

**Table A1.** Part of the indoor fire emergency evacuation knowledge graph object and attribute nodes.

| Object Type | Object | | Attribute |
|---|---|---|---|
| Fire Object | Combustion | Initial Fuel | Volume, Combustion Performance |
| | | Ignition Source | Type |
| | | Accelerant | Adequacy |
| | | Immediate Cause | — — |
| | Loss | Spread Fuel | Volume, Combustion Performance |
| | | Fire Involved Area | Area |
| | | Property Loss | Amount |
| | | Casualty | Death Toll, Number of Injured |
| | Product | Smoke | Toxicity, Visibility, Temperature, Height |
| | | Heat | HRR |
| | | Gas | Composition |

**Table A1.** *Cont.*

| Object Type | Object | | Attribute |
|---|---|---|---|
| Building Object | Building | Fire Building | Area Covered, Number of Stories, Total Area, Structure, Purpose |
| | | Affected Building | |
| | Storey | Fire Storey | Purpose, Area |
| | | Evacuation Storey | |
| | Evacuation Option | Exit | Type, Width, Height, Orientation, State |
| | | Stair | Type, Width, Height, State |
| | | Window | |
| | Fire Facility | Fire Facility | State |
| | | Normal Evacuation Facility | |
| | | Barrier-free Evacuation Facility | |
| Personnel Object | Evacuee | Totally | Number, State |
| | | Elderly | Proportion, State |
| | | Children | |
| | | Disabled | |
| | Emergency Personnel | Fire Fighting Personnel | Number |
| | | Evacuation Guide Personnel | |
| | | Rescue Personnel | |
| | | Communication Personnel | |
| | | Engineering Personnel | |
| | Action | Evacuated | Condition |
| | | Trapped | |
| Time Object | Time Point | Ignition Time | Date, Timestamp |
| | | Detecting Time | |
| | | Alarm Time | |
| | | Attendance Time | |
| | | Extinguishing Time | |
| | | Construction Completion Time | |
| | Time Duration | Combustion Duration | Hours, Minutes, Seconds |
| | | Evacuation Duration | |
| | | I2D Duration | |
| | | D2A Duration | |
| Space Object | Location | Building Location | Province, City, County, Street, Village, Building Number |
| | | Ignition Location | —— |
| | | Evacuee Location | |
| | | Fire Facility Location | |
| | | Evacuation Option Location | |
| | Routine | Fire Spread Routine | Starting Point, End Point, Passby Point |
| | | Evacuation Routine | |

**Table A2.** Part of the State attribute nodes.

| Object | | State |
|---|---|---|
| Evacuation Option | Exit | Blocked by Smoke, Blocked by Items, Unblocked, Unqualified |
| | Stair | |
| | Window | |
| Fire Facility | Hydrant System | Equipped and Operative, Equipped but Inoperative, Unequipped |
| | Fire Extinguisher | |
| | Smoke Control and Exhaust System | |
| | Fire Compartment | |
| | Exit | |
| | Stair | |
| | Fire Elevator | |
| | Fire Alarm System | |
| | Emergency Lighting System | |
| | Evacuation Indicatory Sign | |
| | Accessible Evacuation Ramp | |
| | Barrier-free Evacuation Railings and Handrails | |
| | Barrier-free Evacuation Door | |
| | Barrier-free Evacuation Elevator | |
| | Intelligent Barrier-free Evacuation Equipment | |
| | Emergency Stair-walking Device | |
| | Evacuation Shelter | |
| Evacuee | Totally | Comatose, Sleeping, Drunken, Normal |
| | Elderly | |
| | Children | |
| | Disabled | |

**Table A3.** Part of the relations between indoor fire emergency evacuation knowledge graph nodes.

| Relation Type | Relation | Description |
|---|---|---|
| Whole-part | Contains | B is part of A |
| Inheritance | isA | B is a kind of A |
| Action | hasAction | The relation between Evacuee and Action |
| | Use | The relation between Personnel and Fire Facility |
| | Pass | The relation between Personnel and Component |
| | Extinguish | The relation between Personnel/Fire Facility and Fire |
| | Rescue | The relation between Rescue Personnel and Evacuee |
| | Guide | The relation between Evacuation Guide and Evacuee |
| | Communicate | The relation between Communication Personnel and others |
| | Equip | The relation between Container and Facility |
| | Alarm | The relation between Fire Alarm System and Personnel |
| | Ignite | The relation between Ignition Source and Initial Fuel |
| Temporal | hasTime | The relation between entity and Time Point |
| | Last | The relation between entity and Time Duration |
| Spatial | Locate | The relation between entity and location |
| | farFrom | The relation between Locations |
| | closeTo | The relation between Locations |

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
