# Peer review of "Knowledge Graph Construction to Facilitate Indoor Fire Emergency Evacuation"

_ijgi, doi:10.3390/ijgi12100403_

Round 1

Reviewer 1 Report

Please see the attached file below.

Extensive editing of English language is required.

Reviewer 2 Report

Dear Authors, this manuscript is well designed and the content well explained. The topic of your research is very interesting, because integrates big data analysis and spatial analysis, to generate a knowledge graph, useful to find accessibility solutions, and to avoid disasters.

Below I underlined some corrections and suggestions:

- Lines 70-71. Please add some refererences related to possible 

evacuation scenes and explain why this construction has 

a strong development potential

- Line 133. Please add some refererences related to the management 

of mass data and the data conversion simplification thorugh the use of KG

- Line 153. Please add some refererences regarding the mentioned studies.

lines 153-155. Please add some refererences that confirm this thesis.

line 160. Please explain this concept in a clearer way.

lines 379-382- Please explain the concept in a clearer way, dividing the sentence in multiple parts.

Figure 7- Please use a legend to explain better the content of the figure. It is not clear.

Your english is fine, needs only minor checks.

Reviewer 3 Report

The paper studied the knowledge graph construction to facilitate indoor fire emergency evacuation. Especially, the authors tested the practicability of the proposed method in the case of "5·25" fire accident in Pingdingshan, Henan, China. However, there are some comments on this paper.

1. Actually, the evacation is related with many detailed parameters, especially the fire development process. However, the current proposed parameters is very limited, it is hard to describe the fire development, such as the HRR, smoke toxicity, visibility, etc, which can siginificantly affect the evacuation.

2. The  knowledge graph is only verified by one case, it is not enough, more cases should be added.

It is OK.

Round 2

Reviewer 3 Report

Accept

It's OK.